# A High Temperature Environment Regulates the Olive Oil Biosynthesis Network

**DOI:** 10.3390/plants9091135

**Published:** 2020-09-01

**Authors:** Yael Nissim, Maya Shlosberg, Iris Biton, Yair Many, Adi Doron-Faigenboim, Ran Hovav, Zohar Kerem, Benjamin Avidan, Giora Ben-Ari

**Affiliations:** 1Institute of Plant Sciences, ARO, The Volcani Center, Rishon LeZion 7528809, Israel; yaelns@volcani.agri.gov.il (Y.N.); Maya.shlosberg@mail.huji.ac.il (M.S.); ivrb28@volcani.agri.gov.il (I.B.); yairm@volcani.agri.gov.il (Y.M.); adif@volcani.agri.gov.il (A.D.-F.); ranh@volcani.agri.gov.il (R.H.); vhavidan@volcani.agri.gov.il (B.A.); 2Institute of Biochemistry, Food Science and Nutrition, The Robert H. Smith Faculty of Agriculture, Food and Environment, The Hebrew University of Jerusalem, Rehovot 76100, Israel; zohar.kerem@mail.huji.ac.il

**Keywords:** high temperature, olive, oil, gene expression, biosynthesis

## Abstract

Climate change has been shown to have a substantial impact on agriculture and high temperatures and heat stress are known to have many negative effects on the vegetative and reproductive phases of plants. In a previous study, we addressed the effects of high temperature environments on olive oil yield and quality, by comparing the fruit development and oil accumulation and quality of five olive cultivars placed in high temperature and moderate temperature environments. The aim of the current study was to explore the molecular mechanism resulting in the negative effect of a high temperature environment on oil quantity and quality. We analyzed the transcriptome of two extreme cultivars, ‘Barnea’, which is tolerant to high temperatures in regard to quantity of oil production, but sensitive regarding its quality, and ‘Souri’, which is heat sensitive regarding quantity of oil produced, but relatively tolerant regarding its quality. Transcriptome analyses have been carried out at three different time points during fruit development, focusing on the genes involved in the oil biosynthesis pathway. We found that heat-shock protein expression was induced by the high temperature environment, but the degree of induction was cultivar dependent. The ‘Barnea’ cultivar, whose oil production showed greater tolerance to high temperatures, exhibited a larger degree of induction than the heat sensitive ‘Souri’. On the other hand, many genes involved in olive oil biosynthesis were found to be repressed as a response to high temperatures. *OePDCT* as well as *OeFAD2* genes showed cultivar dependent expression patterns according to their heat tolerance characteristics. The transcription factors *OeDof4.3*, *OeWRI1.1*, *OeDof4.4* and *OeWRI1.2* were identified as key factors in regulating the oil biosynthesis pathway in response to heat stress, based on their co-expression characteristics with other genes involved in this pathway. Our results may contribute to identifying or developing a more heat tolerant cultivar, which will be able to produce high yield and quality oil in a future characterized by global warming.

## 1. Introduction

Abiotic stresses, including high temperatures have a substantial negative impact on the reproductive phase of the plant. Oilseed crops are also negatively affected by heat stress, which has been shown to reduce starch, protein and oil content [1,2]. Plants exposed to temperatures above their optimal growing temperatures, exhibit cellular and metabolic responses which enable the plants to survive [3,4,5,6,7]. The level of damage to crops caused by high temperatures depends on many parameters. The reproductive phase of plant development is more sensitive to high temperatures than the vegetative phase, causing a reduction in yield [8]. Yield reduction in response to heat stress or high temperatures has been reported in wheat, peanut, rice bean and tomato [9,10,11,12,13]. Olive oil accumulation in the mesocarp cells of the fruit is influenced by cultivar type and climactic conditions. It occurs mainly during the summer and decreases during fruit ripening in the fall [14,15].

Olive oil quality is characterized by its sensorial and nutritional properties. Its health benefits are due to both its fatty acid composition, especially oleic acid concentration and minor compounds such as polyphenols. These are strongly affected by factors such as cultivar type, degree of ripeness, fruit load, and soil quality [16]. Several studies have examined the effect of high temperatures on olive oil accumulation [17,18,19,20] and olive oil fatty acid composition [17,19,21,22,23,24]. Oleic acid concentration in olive oil was found to be negatively affected by a high temperature environment [17].

### 1.1. Heat-Shock Proteins

Plants interact with climatic factors such as heat stress by triggering certain mechanisms of defense such as a specific gene expression program. A response to heat stress on the molecular level is found in all living organisms and results in an increase in the induction and synthesis of a group of proteins called heat-shock proteins [25]. Heat-shock proteins positively regulate antioxidant enzymes which detoxify reactive oxygen species. They also enhance plant immunity by the accumulation and stability of pathogenesis-related proteins produced under biotic stresses [26]. During the onset of stress, plants reduce normal protein production, and transcribe and translate heat-shock proteins. Heat-shock proteins have been reported in a wide range of organisms and are highly conserved [26]. Based on molecular weight, HSPs are generally classified into the following sub-families: HSP100, HSP90, HSP70, HSP60, and small HSPs. Several studies have indicated that many high molecular weight HSPs exhibited a response to high-temperature stress [26]. Among the many heat-shock proteins, HSP70s and HSP60s whose response to heat stress has been most intensively studied, have been shown to maintain proper folding under conditions of heat stress with the aid of ATPs [26]. Hsp70s are a class of highly conserved proteins which act as molecular chaperones and play a crucial role in protecting the plant cells from the harmful effects of heat stress. Hsp70 is known to accumulate in heat-stressed tissue and overexpression of Hsp70 was shown to enhance tolerance to heat stress in several plant species including brassica, tobacco and rice [27,28,29,30].

### 1.2. Olive Oil Biosynthesis

Fatty acid biosynthesis begins with Acetyl-CoA carboxylase (ACCase) catalyzing the formation of malonyl-CoA from acetyl-CoA by the ACCase enzyme, which is a complex consisting of three separate proteins or domains: BC, CT and BCCP. ACCase has two isoforms, the prokaryotic enzyme is a heteromeric multisubunit complex (htACC). In eukaryotic cells, the enzyme takes the form of a homomeric multidomain polypeptide (hmACC). The hmACC is a single multifunctional polypeptide, with all three domains forming part of a single polypeptide (ACC1). htACC consists of four separate proteins, BC, BCCP, and CT, which is a heterodimer with α and β subunits. The majority of plants require both types of ACCase, localizing htACC in plastids and hmACC in the cytosol [31]. The MCAT enzyme catalyzes formation of malonyl-ACP from malonyl-CoA. Fatty acids are subsequently produced by a complex of six enzymes referred to as fatty acid synthase (FAS). This complex includes KAS I, KASII and KASIII, KAR, HAD and ENR enzymes. The first step is the condensation of malonyl-ACP with acetyl- CoA by the action of KAS III to produce acetoacetyl-ACP in the plastid. This subsequently forms a β-hydroxyacyl derivative by KAR, which is then dehydrated by HAD and finally reduced by ENR, resulting in a four-carbon acyl-ACP derivative. This forms 3-Katoacyle-ACP by the action of KAS I which can undergo five more chain elongation cycles. The final product of these cycles is palmitoyl-ACP with 16 carbons (C16:0-ACP). Finally, this compound can be elongated to form stearoyl-ACP (C18:0) by KAS II. The most abundant fatty acid in olive oil, oleic acid, is formed by SAD enzyme. The release of the acyl segment from the ACP derivatives is performed by FATA and FATB. These thioesterases show different substrate specificities. FATA preferentially hydrolyzes the 18:1 acyl-ACP, whereas the FATB type is most active with 16:0 and 18:0 acyl-ACPs. The process is localized on the endoplasmic reticulum (ER) by a reaction catalyzed by an oleate desaturase (FAD2) to form linoleic acid (18:2) from oleic acid 18:1. Further desaturation may occur, catalyzed by FAD3 to produce α-linolenic acid (18:3). The assembly of triacylglycerol (TAG) from glycerol-3-phosphate and fatty acids take place on the ER, by the Kornberg–Pricer acylation reactions followed by the Kennedy pathway mediated by the enzymes GPDH, GPAT, LPAAT and PP to form diacylglycerol (DAG). Finally, DAG is an important precursor for synthesis of phospholipids such as PC or storage lipids such as TAG. In order to form TAG, DAG is acylated at the third position by a diacylglycerol acyltransferases (DGATs) or by a phospholipid; diacylglycerol acyltransferase (PDAT) [32]. Another pathway for the DAG is through reactions of phosphatidylcholine diacylglycerol cholinephosphotransferase (PDCT), in which the acyl groups enter PC and can then return to DAG after they are desaturated or otherwise modified on PC [33].

### 1.3. Transcription Factors Regulate Olive Oil Biosynthesis

Regulation of oil biosynthesis by transcription factors (TFs) has been widely investigated and many transcription factors have been reported to affect TAG production [34]. *Leafy cotyledon 1* (*LEC1*) and *LEC1-LIKE* (*L1L*) TFs have been reported to regulate fatty acid synthesis through ACCase in *Arabidopsis*, maize and bean. *DOF4* and *DOF11* TFs were found to regulate fatty acid synthesis through ACCase, *ENR* and *KASII* in *Arabidopsis* and soybean. *MYB73* was reported to regulate fatty acid synthesis through *FATB* and KAS genes in *Arabidopsis* and *Camelina sativa* [34]. However, *WRINKLED1* (*WRI1*) transcription factor is vital to synthesis, and is the “master regulator” of the transcriptional control of the plant oil biosynthesis pathway [35]. Several other TFs were reported to control fatty acid biosynthesis not directly through the biosynthesis genes, but through other genes or transcription factors. These TFs include *LEC2*, *FUS3*, *ABI3*, several bZIP TFs, *MYB96*, *MYB30*, *PICKLE*, *VAL1*, *VAL2*, *ASIL1* and *AP2* [34].

We have lately addressed the effect of high temperature environments on olive oil yield and quality, by comparing the fruit development and oil accumulation and quality of five olive cultivars placed in high temperature (HT) and low temperature (MT) environments [36]. We found that the effects of a high temperature environment are genotype dependent and in general, high temperatures during fruit development decreased oil accumulation and quality. ‘Barnea’ cultivar final oil content was not affected by the high temperature environment, but ‘Souri’ cultivar exhibited a dramatic decrease of 8% in oil concentration calculated as a percentage of dry fruit weight at harvest in response to high ambient temperatures. Regarding the quality of oil produced, the ‘Souri’ cultivar proved more tolerant to a high temperature environment than any other of the tested cultivars. The ‘Barnea’ oil had a polyphenol level of 404 mg/g oil in the MT environment and only 156 mg/g oil in the HT environment. In addition, the ‘Barnea’ oil showed a decrease of 4.8% in oleic acid concentration of oil extracted from fruit grown in HT conditions compared to the MT environment. The ‘Souri’ cultivar oil contained high polyphenol levels of 905 and 772 mg/g oil in the MT and HT environments respectively. In addition, the ‘Souri’ oil showed a decrease of only 3.7% in oleic acid concentration of oil extracted from fruit grown under HT conditions compared to oil from MT fruit. The aim of the current study was to explore the molecular mechanism underlying the negative effect of the high temperature environment on oil quantity and quality. In order to identify the molecular mechanism and the possible blocks, in which HT negatively affects oil accumulation and quality, we analyzed the transcriptome of the two extreme cultivars, ‘Barnea’ and ‘Souri’. Transcriptome analyses were carried out at three different time points during fruit development, focusing on the genes involved in the oil biosynthesis pathway.

## 2. Materials and Methods

### 2.1. Experimental Design and Plant Material

The olive cultivars ‘Barnea’, ‘Coratina’, ‘Koroneiki’, ‘Souri’ and ‘Picholine’ were used in this study. Of these, ‘Barnea’, which is tolerant to HT environment, regarding fruit development and oil accumulation and ‘Souri’, tolerant to HT environment, regarding oil quality, were used for transcriptome analysis. Twelve olive trees in 50 L pots of each of the five cultivars were placed in an extremely hot climate zone at Tirat Zvi village (HT environment) and in a more temperate environment (MT environment) during 2016 and 2017. Experiment design and all measurements were described previously in detail [36]. During 2017, the mesocarp of five fruits from each of the six trees was collected and three replicates of mesocarp from a mix of ten fruits from two trees each were used for transcriptome analysis. Mesocarp was sampled at three time points during the season, at 83, 104 and 146 days post anthesis (DPA).

### 2.2. Transcriptome Analysis

Fruits were removed from the tree, the mesocarp of a pool of 10 fruits from two trees was excised, mixed in a tube and immediately frozen in liquid nitrogen. Total RNA was extracted from olive mesocarp tissue using Sigma (Sigma Aldrich, Darmstadt, Germany) plant total RNA extraction kit according to the manufacturer’s instructions. DNA was removed by treatment with RNAase-free DNAase at 37 °C for 30 min. Total RNA samples were used to prepare 24 cDNA libraries of the two cultivars (‘Barnea’ and ‘Souri’) at three time points during fruit development (83, 104 and 146 DPA) in two locations (HT and MT environments) and two biological replicates, using Illumina’s TruSeq RNA library preparation kit according to the manufacturer’s instructions. cDNA libraries were sequenced on NextSeq high output 75 cycles. Sequencing was performed by the Sequencing unit of the Crown Institute for Genomics in the Weizmann Institute of Science, Israel. Raw reads were subjected to a filtering and cleaning procedure using the FASTX Toolkit (http://hannonlab.cshl.edu/fastx_toolkit/, version 0.0.13.2) for: (1) trimming read-end nucleotides with quality scores <30 using fastq_quality_trimmer; (2) removing reads with less than 70% base pairs with quality score ≤30 using fastq_quality_filter. Cleaned reads, obtained after processing, were assembled de novo using Trinity software (GitHub Inc., San Francisco, CA, USA) [37] with trimmomaticSE option to remove adaptors [38]. Only transcripts with a minimum length of 200 bp were analyzed against OE6 transcriptome reference. The transcript quantification (the number of reads per gene) from RNA-Seq data was performed using bowtie2 aligner [39] and the Expectation-Maximization method (RSEM) [40]

### 2.3. RT-PCR

Specific primers were designed using Primer3 software (Elixir, Narva maantee, Estonia) [41]. Primers for the various genes were designed to amplify a region containing more than one exon, so that genomic DNA would not be amplified. A total of 2.5 µg RNA from each sample was treated with RQ DNase (Promega, Madison, WI, USA) and reverse-transcribed using random hexamer primers (Promega). Real-time PCR was carried out using the SYBR green amplification kit (ABgene, Blenheim Road, Epsom, UK) according to manufacturer’s instructions. Each reaction contained 1 µL cDNA and 1 µM of each primer from the relevant primer pair in a final volume of 10 µL. Quantification of real-time PCR products was carried out by detection of SYBR green fluorescence on a StepOnePlus™ system (Applied Biosystems, Foster City, CA, USA). Dilution series of cDNA were created for each set of and a standard curve was established for each gene. Triplicate of cDNA were used and each reaction was subjected to melting-point analysis to confirm single amplified products. At least two biological repeats were carried out for each gene. Transcript levels were estimated using a standard curve for each gene, and these levels were normalized against the amount of *OeACTIN* transcript level in each sample. The sequences of the primers used are listed in the Appendix A.

### 2.4. Data Analysis

Differential expression analysis was done with edgeR package (Bioconductor, Buffalo, NY, USA) [42], transcripts that were more than two fold differentially expressed with false discovery corrected statistical significance of at most 0.01 were considered differentially expressed [43]. Hierarchical clustering of gene expression and visualization of heat map were performed using R Bioconductor (Bioconductor, Buffalo, NY, USA) [44], using the normalized log-FPKM values (Z score) of each gene. Venn diagrams were constructed using the online Venny 2.0 software (http://bioinfogp.cnb.csic.es/tools/venny/). From each gene family participating in the oil biosynthesis pathway, we presented only genes that showed a normalized expression level of at least 200 FPKM in at least one time-point and environment, assuming that genes with low expression levels are not involved in the biosynthesis pathway. Co-expression network analysis inferred by using Conet plugin of cytoscape software (Cytoscape.js, Toronto, ON, Canada) that was based on a Pearson correlation (cut-off of r > 0.75).

### 2.5. Statistical Analysis

RT-PCR statistical analyses were conducted using JMP software (SAS, Cary, NC, USA) [45]. The RT-PCR results were subjected to two-way analysis of variance (ANOVA) including full factorial analysis, for their dependence on the two independent variables of tree location and cultivar type, including the interaction between them. Since we encountered significant interaction between the two factors in all gene expression analysis, a Tukey-Kramer test, based on multiple comparison correction was performed, in order to compare the various levels of interaction.

## 3. Results

### 3.1. Gene Expression Regulation

In order to understand the mechanism in which high temperature summers affect olive oil biosynthesis we used transcriptome analysis. Two out of five analyzed [36] olive cultivars were chosen, ‘Barnea’ and ‘Souri’. ‘Barnea’ exhibited tolerance to HT environment in term of oil concentration at harvest, but its oil quality dramatically decreased in an HT environment. ‘Souri’ exhibited sensitivity to HT environment in term of oil concentration at harvest, but proved more tolerant to HT in oil quality, compared to all the tested cultivars. Mesocarp transcriptomic of the two cultivars was analyzed at three time points during fruit development in HT and MT environments, 83, 104 and 146 days post anthesis (DPA).

The transcriptome sequencing resulted in an average of 21,748,879 reads for each sequenced sample. Among them, 96.54% were clean reads and 83.47% of the reads were significantly mapped to the olive genome [46] (Appendix A). The sequencing data were deposited in the NCBI Sequence Read Archive (SRA) database as bioproject PRJNA638790.

Comparing expression levels of each gene in the HT and MT environments for each cultivar and each time point, identified the regulated genes in response to different environments. The ‘Barnea’ cultivar had 2437, 2795 and 3265 genes that were significantly upregulated in the HT environment at 83, 104 and 146 DPA respectively. The ‘Souri’ cultivar had 3009, 2625 and 3129 genes that were significantly upregulated in the HT environment at 83, 104 and 146 DPA respectively. The ‘Barnea’ cultivar had 2500, 2634 and 3473 genes that were significantly upregulated in the MT environment at 83, 104 and 146 DPA respectively. The ‘Souri’ cultivar had 2953, 3053 and 3174 genes that were significantly upregulated in the MT environment at 83, 104 and 146 DPA respectively. Among the differentially expressed genes, more genes were up-regulated at all three time points (1193, 1226, 1291 and 1362 for ‘Barnea’-MT-Up, ‘Barnea’-HT-Up, ‘Souri’-MT-Up and ‘Souri’-HT-Up respectively) compared to those that were unique for specific time point or common to only two time points (Appendix A). Comparing the identity of the regulated genes between the three selected time points showed that the highest number of regulated genes is common at all three time points. However, when comparing the regulated genes between the two cultivars, we found that the number of regulated genes that were cultivar specific in each of the two cultivars was similar to the number of regulated genes common to both cultivars, in all three time points (Appendix A).

### 3.2. Heat-Shock Protein Genes are Upregulated in the HT Environment

Hierarchical clustering of all 110 heat-shock protein genes revealed that the gene expression profile of the heat-shock protein genes in ‘Barnea’ and ‘Souri’ in all three time points in the MT environment is similar (Figure 1). Most of these genes were expressed at low rates in both cultivars and at all three time points. In addition, The MT samples were separated by cultivar into two cluster groups. For the ‘Souri’, three time points in MT were clustered in one group whereas for ‘Barnea’, three time points in MT were clustered in another group. In the HT samples, ‘Souri’ at 104 DPA and ‘Barnea’ at 83 DPA clustered together and most of the heat-shock protein genes were expressed at low rates in these samples. ‘Souri’ at 83 and 146 DPA and ‘Barnea’ at 104 DPA were clustered in one group and most of the heat-shock protein genes were highly expressed. Above all, the heat-shock protein genes of ‘Barnea’ at 146 DPA in the HT environment were expressed at extremely high rates. Validation of the heat-shock protein gene expression pattern in HT and MT environments was carried out by RT-PCR analysis of the expression pattern of *OeHSP70* known to be induced by heat shock. 

Expression level of *OeHSP70* (*OE6A062772*) at 146 DPA, in all five cultivars [36] was analyzed (Appendix A). *OeHSP70* expression level in the MT environment in all five cultivars was similar and relatively low compared to its expression in the HT environment. The expression level of *OeHSP70* in the HT environment varied between the cultivars. The ‘Souri’ cultivar showed the lowest expression, ‘Picholine’ and ‘Coratina’ exhibited moderate expression, and ‘Barnea’ and ‘Koroneiki’ showed the highest expression, in which, ‘Barnea’ *OeHSP70* expression was found to be 8.7 times greater than *OeHSP70* expression in the ‘Souri’ cultivar in the HT environment.

### 3.3. The Effect of High Summer Temperatures on the Olive Oil Biosynthesis Pathway

The biochemical pathways leading to oil (triacylglycerol) synthesis in the olive fruit mesocarp involve many subcellular organelles and multiple enzymes, beginning with the precursor, acetyl-CoA and concluding with triacylglycerol (TAG). Based on the transcriptome analysis of ‘Barnea’ and ‘Souri’ at the two contrasting environments at three time points during fruit development, we focused on the gene expression pattern of all genes involved in olive oil biosynthesis (Figure 2). Figure 2 presents the expression pattern of the ‘Souri’ and the ‘Barnea’ cultivars. The following stages describe in detail oil synthesis in the ‘Souri’, a cultivar we have shown to be sensitive to a high temperature environment as reflected by oil biosynthesis [36]. The first step of the fatty acids biosynthesis is catalyzing the formation of malonyl- CoA from acetyl-CoA by Acetyl-CoA carboxylase (ACCase). Gene expression pattern of the genes involved in the multienzyme complex was similar in the HT and MT environments beside *OeBCCP1* (*OE6A092496*), which was induced in the HT environment. On the other hand, *OeACC1* (*OE6A049983*) was expressed three to six fold (depending on the sample date) higher in the MT environment compared to the HT. *OeMCAT*, which catalyzed malonyl-CoA was expressed similarly in both environments. Fatty acid synthase (FAS) is an easily dissociated multisubunit complex consisting of the monofunctional enzymes KASI, KASII, KASIII, KAR, HAD and ENR. *OeHAD*, *OeENR* and *OeKAR* were expressed similarly in the HT and MT environments at the first sample date (83 DPA), at which *OeKAR* expression was higher in the MT compared to the HT environment. However, *OeKASI*, *OeKASII* and *OeKASIII* expression was higher in the MT compared to the HT environment in all three sample dates. Both stearoyl-ACP Δ9-desaturase (SAD) family genes (*OE6A020845* and *OE6A118450*) were very highly expressed in both environments. The last stage of de novo fatty acid synthesis is regulated by the acyl-ACP thioesterases, FATA and FATB. The two FAT genes, *OeFATA* and *OeFATB* were induced in the MT environment in comparison to their expression in the HT. This is especially true for the *OeFATB* gene (*OE6A029754*), which is expressed at a threefold level in the MT environment compared to its expression in the HT at all three time points. The modification of oleic acid (18:1) into linoleic acid (18:2) and linolenic acid (18:3) is catalyzed by the fatty acid desaturases FAD2 and FAD3 respectively in the Endoplasmic Reticulum (ER). OeFAD3 genes were not expressed in our samples. However, OeFAD2 genes showed higher expression in the HT compared to the MT environment, especially regarding *OeFAD2-5* (*OE6A098403*) at the last sample date (146 DPA) and *OeFAD2-1* (*OE6A069627*) at the first sample date (83 DPA). The expression level of *OeFAD2-5* in the ‘Souri’ cultivar at 146 DPA was found to be 11,700 FPKM in the HT environment, significantly higher than the 1976 FPKM level found in the MT environment. Assembly of fatty acids onto the glycerol backbone to create triacylglycerol (TAG) begins with a series of reactions with dihydroxyacetone phosphate (DHAP) as a precursor and diacylglycerol (DAG) as the product with four enzymes involved (GPDH, GPAT, LPAAT and PP). The expression level of the main genes involved in this reaction, *OeGPDH* (*OE6A105859*) and *OeGPAT* (*OE6A109037*) was significantly higher in the MT environment compared to that in the HT. However, *OeLPAAT* gene family consists of one gene (*OE6A051077*) that was induced in the HT environment and another single gene (*OE6A111035*) significantly induced in the MT environment. *OePP* genes were marginally expressed in both environments at all three time points.

DAG induction can proceed along three different pathways, two of them with TAG as the final product, and one, regulated by PDCT, to return to the phosphatidylcholine pool. DAG can create TAG by the reaction of PDAT or by the reaction of the two DAG proteins, DGAT1 and DGAT2. 

The *OePDAT* gene family consists of three genes that were highly expressed in our study. *OE6A007392* had a significantly higher expression level in the MT compared to the HT environment at all three time points, whereas *OE6A078092* and *OE6A031530* had a significantly higher expression level in the HT compare to the MT. *OeDGAT1* (*OE6A093626*) was significantly expressed at a higher level in the MT compare to the HT environment, especially at 83 DPA, whereas *OeDGAT2* (*OE6A030092*) expression was higher in the MT only at the latter two time points at 104 and 146 DPA. The two expressed *OePDCT* genes (*OE6A047231* and *OE6A061761*) showed higher expression levels in the HT compare to the MT environment at the time points 104 and 146 DPA.

The ‘Barnea’ gene expression pattern was similar to that of the ‘Souri’ with the same trends between the two environments (Figure 2). The exceptional genes were FATA, in which one, *OeFATA* (*OE6A018491*), was highly expressed in both environments at all three time points. The expression pattern of two *FAD2* genes, *OeFAD2-5* and *Oe-FAD2-1* was also different in the ‘Barnea’ cultivar compared to the ‘Souri’. The ‘Barnea’ cultivar showed extremely high expression of *OeFAD2-5* in the HT environment at all three time points and relatively high expression in the MT environment, especially at 146 DPA. *Oe-FAD2-1* was very highly expressed in the first two sample dates (83 and 104 DPA) in the HT environment and moderately expressed in the MT environment at all time-points. The ‘Barnea’ cultivar showed the same trend in the expression of *OeDGAT1* and *OeDGAT2* in both environments as the ‘Souri’ cultivar, only in the ‘Barnea’ cultivar, the differences between the expression levels in the MT and HT environments were smaller. Unlike the ‘Souri’ cultivar, the ‘Barnea’ *OePDCT* genes were expressed similarly in both environments.

Validation of the oil biosynthesis gene expression pattern in HT and MT environment was carried out by RT-PCR analysis of the expression pattern of *OeACC1* and *OeFAD2-1* at 146 DPA, in all five cultivars [36] (Figure 3). 

*OeACC1* expression was significantly higher in the MT compared to the HT environment in all five cultivars, with the greatest differences between the two environments in the s ‘Koroneiki’ and ‘Souri’ cultivars. *OeFAD2-1* expression level was relatively high in the MT environment in ‘Barnea’ and ‘Picholine’ cultivars and low in the other three. In the HT environment, *OeFAD2-1* expression level in the ‘Koroneiki’ cultivar was low, and in the ‘Souri’ cultivar it was no expressed at all. In ‘Coratina’, it was expressed at a relatively high level and in the ‘Barnea’ and ‘Picholine’ cultivars was induced at extremely high levels.

### 3.4. Transcription Factors Regulating Olive Oil Biosynthesis Pathway

We next explored the regulatory roles of transcription factors in the oil biosynthesis pathway. In order to identify putative targets of the transcription factors known to directly regulate the oil biosynthesis pathway genes, we used co-expression analysis to identify significant and high (r > 0.75) expression pattern correlation between pairs of transcription factors and genes involved in the oil biosynthesis pathway (Figure 4). We identified eight WRI1 genes in the olive genome (*OE6A039756*, *OE6A079258*, *OE6A099997*, *OE6A092113*, *OE6A061030*, *OE6A094680*, *OE6A054009* and *OE6A025576*), four Dof4 genes (*OE6A082367*, *OE6A085395*, *OE6A104771* and *OE6A039076*), four LEC1 genes (*OE6A094722*, *OE6A016910*, *OE6A001317* and *OE6A033717*), one Dof11 gene (*OE6A032821*), one L1L gene (*OE6A077959*) and one MYB73 gene (*OE6A056119*). The transcription factors with the highest number of putative target genes, based on co-expression, were *OeDof4*.3, *OeWRI1.1*, *OeDof4.4* and *OeWRI1.2* with 15, 12, 11 and 10 positive co-expressed oil biosynthesis genes. The oil biosynthesis genes with the highest number of putative transcription factors control, based on co-expression, were *OeACC1*, *OeGPDH.3*, *OsDGAT1*, *OeFATB.2* and *OePDAT.3* with 9, 7, 6, 6 and 6 co-expressed transcription factors.

## 4. Discussion

In this study, we addressed the molecular effects of an environment of consistently high temperatures on oil accumulation and composition in different cultivars. We used two cultivars that differ in their response to HT environment. The ‘Barnea’ cultivar, which had been found to be relatively resistant to HT environment in term of oil accumulation, but not in its oil quality and the ‘Souri’ cultivar, found to be sensitive to HT environment in term of oil accumulation, but relatively resistant to HT environment in term of oil quality [36]. We found that during fruit development and during the most critical period of oil accumulation, both cultivars, the heat tolerant ‘Barnea’ cultivar as well as the heat sensitive, ‘Souri’, induced a heat stress protein response in the HT environment. We characterized the gene expression pattern of the genes involved in the olive oil biosynthesis and found that many genes are repressed in response to a high temperature environment. Finally, we characterized the TF-oil biosynthesis network by its response to an MT vs. HT environment in the two analyzed cultivars, ‘Barnea’ and ‘Souri’.

### 4.1. HT Environment Induced A Heat Stress Response in Both Cultivars

Heat stress or high temperature affects the metabolism of plants, especially cell membranes and physiological processes such as photosynthesis, respiration, and water regulation. This defense at the molecular level is very important for survival and growth of plants. Plants show a series of molecular responses to these stresses, among them is the synthesis of heat-shock proteins. Heat-shock proteins act as molecular chaperones regulating the folding, accumulation, localization and degradation, of proteins in plants [25]. Our study focused on heat stress only. We treated the plant with enough water and fertilization to attempt to isolate heat stress from other abiotic stresses [36]. Characterization of the expression pattern of heat-shock proteins in our study, revealed that most of the heat-shock proteins induced dramatically in an HT environment were HSP20-like chaperones, although some genes belonging to the HSP70 family had the same expression pattern. Most heat-shock proteins in our study underwent high expression in the HT environment and low expression in the MT environment at all three time-points (Figure 1). The ‘Souri’ cultivar presents a unique expression pattern of many HSP genes that were induced in 83 DPA, then repressed in 104 DPA, and up-regulated again in 146 DPA. We could not find any reasonable explanation for this gene expression pattern. At 104 and 146 DPA, the ratio between the expression of the heat-shock protein genes in the HT and the MT environment was higher in the ‘Barnea’ compared to the ‘Souri’ cultivar. Assuming that the ‘Barnea’ cultivar is more tolerant to heat stress compared to the ‘Souri’ cultivar in regard to fruit development and oil accumulation [36], suggests that induction of heat-shock proteins may contribute to the olive plant’s tolerance to heat stress. This assumption is in agreement with other studies which found that heat-shock proteins are involved in heat tolerance [26,47,48]. Analysis of *OeHSP70* expression also showed the same pattern, in which the differences between the expression levels in the HT and the MT environments were much higher in the ‘Barnea’ cultivar than in the ‘Souri’. However, contrary to that, the ‘Koroneiki’ cultivar, which is as sensitive to high temperatures as the ‘Souri’ [36], showed a similar pattern of *OeHSP70* expression as the ‘Barnea’ and not, as we might have predicted, to the ‘Souri’ (Appendix A). The *OeHSP70* expression pattern in the ‘Souri’ and ‘Barnea’ cultivars as revealed in the RT-PCR experiment showed the same trend as was seen in the RNA-seq results. However, the ratio between the expression level in HT and MT environments was slightly different. This ratio was 1308:1 and 12:1 in the ‘Barnea’ and ‘Souri’ cultivars respectively according to the RNA-seq results and 630:1 and 101:1 respectively according to the RT-PCR results.

### 4.2. High Summer Temperatures Repress Genes Involved in the Olive Oil Biosynthesis Pathway

Formation of malonyl-CoA from acetyl-CoA is catalyze by ACCase. In our study, both the genes composing the multienzyme complex (htACC) as well as hmACC (*OeACC1*) were expressed in the olive mesocarp (Figure 2 and Figure 3). It has been suggested that in monocotyledons both forms are active, whereas in dicotyledons, such as olives, hmACC alone is active [32]. However, according to the expression levels of both forms, our results suggest that in olives both forms may be active and contribute to the malonyl-CoA, a precursor of fatty acids. Many genes involved in olive oil biosynthesis are not sensitive to warm temperatures during fruit development and show similar expression in both HT and MT environments. These genes include the genes responsible for forming the htACC complex, MCAT, KAR, HAD, ENR, SAD and LPAAT. However, many genes involved in oil biosynthesis are sensitive to a high temperature environment and showed significant lower expression in the HT compare to the MT environment. The high temperature sensitive genes were ACC1, KASI, II and III, FATA and FATB, GPDH, GPAT, DGATs and PDAT. FAD2 and PDCT were also high temperature sensitive genes, but showed increased expression in HT compared to the MT environment. Oil accumulation was shown to be temperature sensitive and high temperatures resulted in decreased olive oil content in the mesocarp [17,36]. Conversion of acetyl-CoA to malonyl-CoA is catalyzed by ACC1 which is a key regulator of fatty acid synthesis. Overexpression of *ACC1* in yeast significantly increased the total fatty acid content [49,50,51]. We found that *OeACC1* (*OE6A049983*) is repressed in an HT environment. At 146 DPA, the ratio of the *OeACC1* expression level between MT and HT environment was 2.9:1 in the heat tolerant ‘Barnea’ cultivar and 24:1 in the heat sensitive cultivar ‘Souri’. However, in the sensitive cultivar ‘Koroneiki’ the ratio was 4.1:1 and in the tolerant cultivars ‘Coratina’ and ‘Picholine’ the ratio was 9:1 and 4.8:1, respectively (Figure 3). Based on these results, we can conclude that *OeACC1* is repressed at high temperatures. However, this gene cannot serve as a marker to discriminate between tolerant and sensitive cultivars in regard to heat stress. All three KAS genes were found to be sensitive to high temperatures and their expression was repressed in the HT environment. Characterizing only the gene with the highest expression from each gene family, we found that KASI (*OE6A049543*) was significantly overexpressed in the MT environment at all three time-points in the ‘Barnea’ cultivar, but only at 146 DPA in the ‘Souri’ cultivar. KASII (*OE6A089647*) was significantly overexpressed in the MT environment at all three time-points in the ‘Souri’ cultivar, but only at 83 and 104 DPA in the ‘Barnea’ cultivar. KASIII (*OE6A034993*) was significantly overexpressed in the MT environment at all three time-points in both cultivars. KAS genes, like *OeACC1*, are repressed in high temperatures. However, they also cannot serve as markers to discriminate between tolerant and sensitive cultivars to heat stress. Most of the genes involved in the fatty acid synthesis occurring in the ER were found to be HT regulated. GPDH, GPAT, DGAT1, DGAT2 and PDAT expression was repressed in HT environment. DGAT plays a key role in determining the carbon flux into TAG [52], and repression of its expression may lead to decreased oil accumulation as we found earlier [36]. However, FAD2 and PDCT were induced in under these conditions (Figure 2). Fatty acid modification from oleic acid to linoleic acid (C18:2) is catalyzed by FAD2 and FAD6 and from oleic acid to linolenic acid (C18:3) by FAD3 [32]. FAD3 genes were not expressed and the OeFAD6 gene was minimally expressed in both cultivars and in both environments. Among the FAD2 genes, *OeFAD2-2* and *OeFAD2-3* were also expressed at low levels in both cultivars and in both environments. *OeFAD2-5*, *OeFAD2-1* and *OeFAD2-4* were highly expressed in some samples. In the ‘Souri’ cultivar, *OeFAD2-4* was expressed similarly in both environments, *OeFAD2-1* expressed similarly in both environments at 104 and 146 DPA but induced in the HT environment at 83 DPA, *OeFAD2-5* induced in the HT environment at 83 and 146 DPA. In the ‘Barnea’ cultivar, with the exception of *OeFAD2-4*, which was expressed similarly in both environments at 146 DPA, all three expressed FAD2 genes were induced in the HT environment at all sampled time-points. This is in agreement with other studies which found that FAD2 genes expression was induced by high temperatures [53]. However, other studies found that low temperatures also induced FAD2 genes in olives [54]. PDCT genes were induced in the HT environment in the ‘Souri’ cultivar, but not in the ‘Barnea’ cultivar. The PDCT enzyme uses the DAG as precursor and returns it to the PC pool, thus reducing the amount of DAG that continues to synthesis of TAG and repressing oil production. The main genes whose expression was cultivar specific regarding ‘Barnea’ and the ‘Souri’, are PDCT and FAD2. PDCT was induced in the HT environment only in the heat sensitive cultivar, ‘Souri’, and it’s activity repressed oil production in that environment. However, in the ‘Barnea’ cultivar, its expression was constant in both environments. FAD2 was highly induced in the HT environment in the heat sensitive cultivar, ‘Barnea’ and its’ effect was evident in the quality of the oil produced under high temperature conditions. It was only slightly induced under similar environmental conditions in the heat resistant cultivar, ‘Souri’. *OeFAD2-1* expression level in all five cultivars in both environments (Figure 3), was found to be in relatively good agreement with the oil quality sensitivity of these cultivars [36]. The cultivars ‘Barnea’ and ‘Picholine’ were found to be the most heat sensitive in terms of oil quality and these cultivars showed the highest expression level of *OeFAD2-1 in the HT environment. The ‘Souri’ cultivar was characterized as the most heat resistant of the five in terms of oil quality and showed no expression of OeFAD2-1* under HT conditions. However, the cultivars ‘Koroneiki’ and ‘Coratina’ showed oil quality sensitivity under high temperature conditions, with relatively low expression of *OeFAD2-1*. The ‘Koroneiki’ cultivar, although found to be heat sensitive in term of oil quality, showed similar expression levels of *OeFAD2-1 in both environments. OeACC1* and *OeFAD2-1* expression patterns in the ‘Souri’ and ‘Barnea’ cultivars as demonstrated in the RT-PCR experiment showed the same trend as in the RNA-seq results. However, the ratio between the expression level in HT and MT environments was slightly different. This ratio in *OeACC1* was 1:2.5 and 1:2 in the ‘Barnea’ and ‘Souri’ cultivars respectively according to the RNA-seq results and 1:2.9 and 1:25 respectively as appeared in the RT-PCR results. For *OeFAD2-1*, this ratio was 2.7:1 and 1:1.5 in the ‘Barnea’ and ‘Souri’ cultivars respectively according to the RNA-seq results and 2.7:1 and 1:10 respectively according to the RT-PCR results.

### 4.3. WRI1 and Dof4 Are the Main Transcription Factors Regulating Olive Oil Biosynthesis Pathway

Network analysis of the transcription factors and oil biosynthesis genes reveals transcription factors hubs, which hypothetically could regulate the genes involved in fatty acid biosynthesis in response to high temperatures. We found that among the transcription factors known to regulate oil biosynthesis, the transcription factors *OeDof4*.3, *OeWRI1.1*, *OeDof4.4* and *OeWRI1.2* were co-expressed with a high number of oil biosynthesis genes. The LEC1, L1L, MYB73 and Dof11 transcription factors were co-expressed with a relatively low number of genes involved in oil biosynthesis, suggesting that these TFs are not the main factors affecting the response of oil production to high temperatures. WRI1 was already suggested as the main regulator in transcriptional control of plant oil biosynthesis [35]. However, in order to validate the role of the suggested hub, more studies need to be carried out, including identifying the binding motifs of the transcription factors in the target gene promoters as well as chromatin immunoprecipitation experiments to show that indeed the suggested transcription factors interact with the promotor genes involved in oil biosynthesis.

## 5. Conclusions

Our study demonstrates the negative effect of a high temperature environment on several key genes critical to the production and quality of olive oil. Different olive cultivars have developed a variety of mechanisms to deal with different aspects of high temperature damage. In order to elucidate the mechanism of high temperature damage to olive oil, we characterized the expression pattern of genes involved in the olive oil biosynthesis pathway in a high temperature (HT) environment compared to expression patterns under moderate conditions (MT). We found that most of the genes regulated by high temperatures are common to different stages of fruit development. However, many of them are cultivar dependent. We hypothesize that a strong induction of heat-shock proteins, especially during the late stage of fruit development, can indicate heat tolerance. Many genes involved in the olive oil biosynthesis are down-regulated as a response to high temperatures. *OePDCT* as well as *OeFAD2* genes showed cultivar dependent expression patterns strongly related to their heat tolerance. Hence, these genes are recommended as markers for screening of various cultivar to test their tolerance level for high summer temperatures. We also found that the transcription factors *OeDof4*.3, *OeWRI1.1*, *OeDof4.4* and *OeWRI1.2* were co-expressed with a high number of olive oil biosynthesis genes and therefore seem to be key factors in regulating the oil biosynthesis pathway in response to heat stress. Due to climate changes in recent years and the forecast for the future, the mechanisms of the various olive cultivars response to heat stress should be characterized in detail in order to identify existing cultivars or to develop new ones tolerant to high summer temperatures. These will hopefully produce high yields as well as quality oil in a changing environment.

## Figures and Tables

**Figure 1 plants-09-01135-f001:**
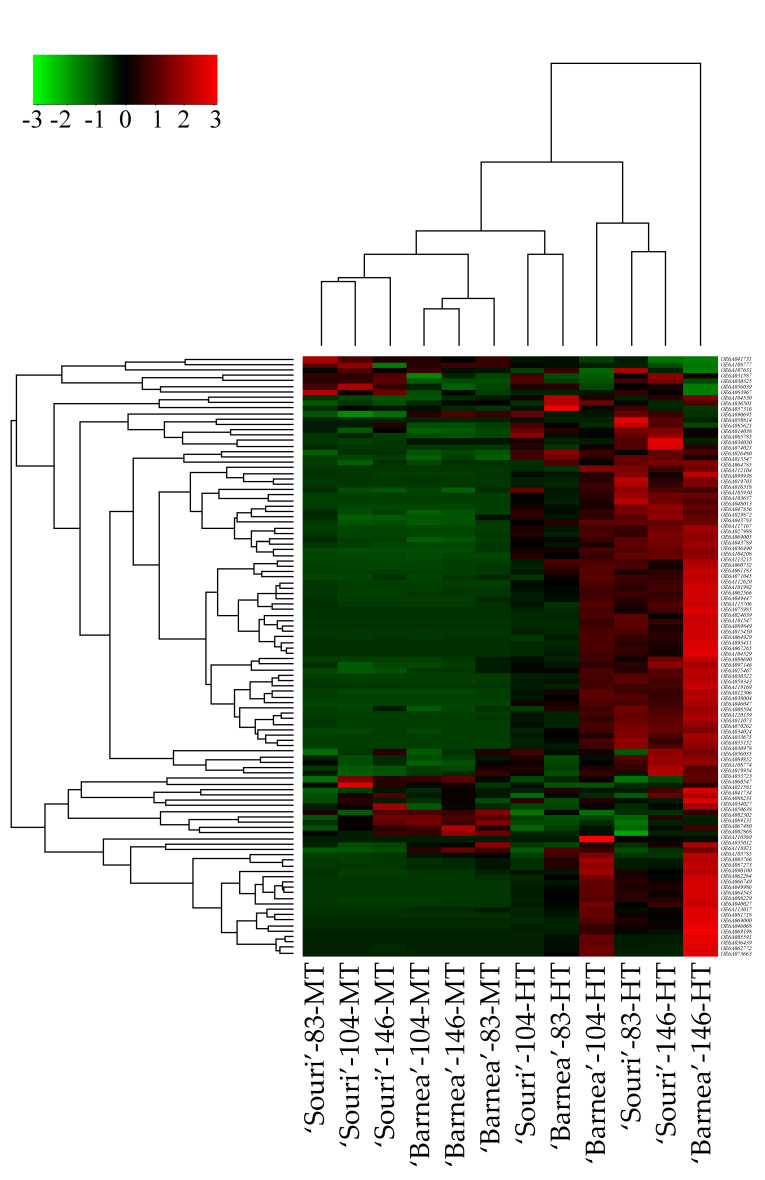
Hierarchical clustering of all heat-shock protein genes. Genes (rows) and samples (columns) were clustered according to their expression pattern. Sample names below include the cultivar-sample DPA-environment. Expression levels are indicated on an abundance scale of green to red.

**Figure 2 plants-09-01135-f002:**
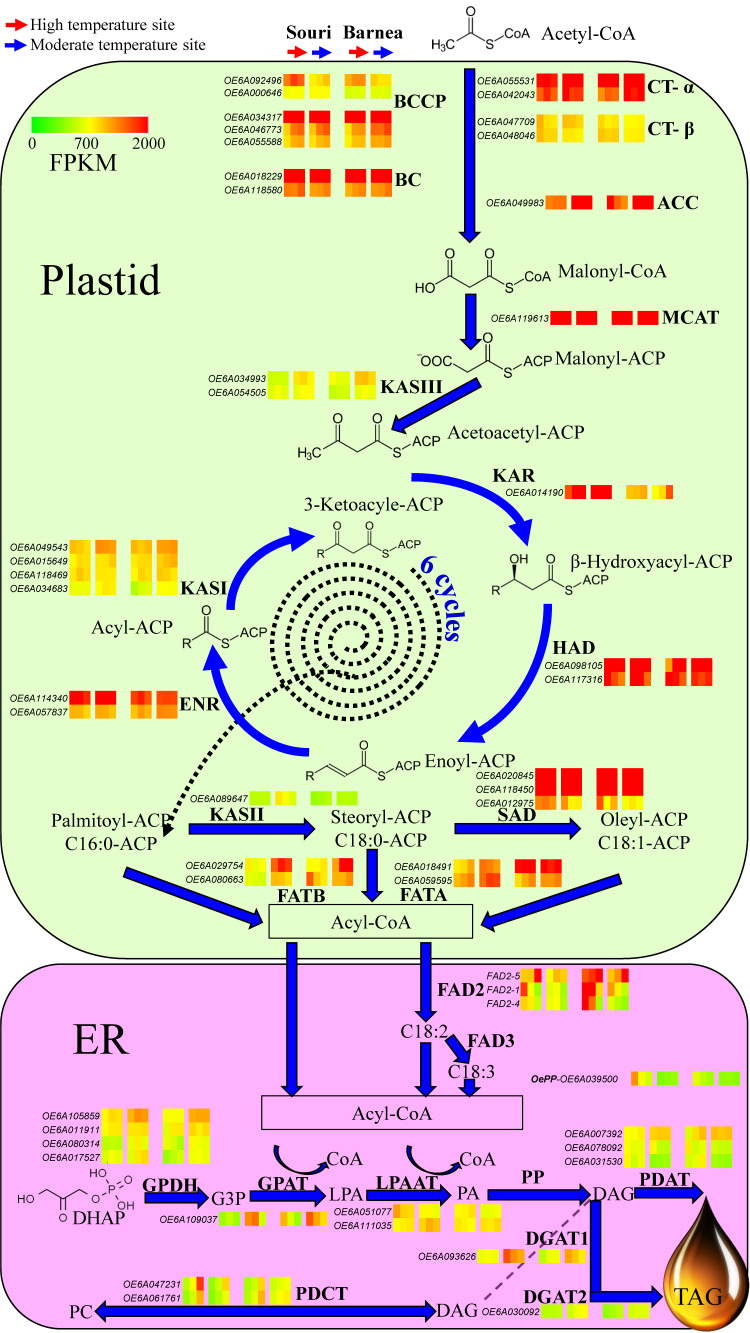
Expression pattern of the genes involved in the olive oil biosynthesis pathway in fruit from the HT site compared to that of fruit grown in the MT environment. Expression levels for each gene are presented in a green to red scale above or beside the name of the enzyme. For each enzyme, only genes with a FPKM of above 200 in at least one time point and environment appears. For each gene, expression level at the three time points sampled at the two environments is presented. In each gene, the six left squares represent the expression pattern in the ‘Souri’ cultivar, whereas the six right squares represent the expression pattern in the ‘Barnea’ cultivar. Within each cultivar, the three left squares represent the expression level at 83, 104 and 146 DPA in the HT environment, whereas the three right squares represent the expression level at the same sample dates in the MT environment. The various cell components appear in different colors. The reaction begins with cytosol (white background), then advances to the plastid (light green background) and terminates at the endoplasmic reticulum (ER; purple background). Abbreviations: *BCCP*—biotin carboxyl carrier protein, *BC*—biotin carboxylase, *CT*—carboxyl transferase, *ACC*—acetyl-CoA carboxylase, *MCAT*—malonyl-CoA:ACP transacylase, *ACP*—acyl carrier protein, *KAS*—β-ketoacyl-ACP synthase, *KAR*—β-ketoacyl-ACP reductase, *HAD*—β-hydroxyacyl-ACP dehydrase, *ENR*—enoyl-ACP reductase, *SAD*—stearoyl-ACP desaturase, *FAT*—fatty acyl-ACP thioesterases, *FAD*—fatty acid desaturases, *GPDH*—glycerol 3-phosphate dehydrogenase, *GPAT*—glycerol 3-phosphate acyltransferase, *LPAAT*—lysophosphatidate acyltransferase, *PP*—phosphatidate phosphohydrolase, *PDCT*—phosphatidylcholine diacylglycerol cholinephosphotransferase, *DGAT*—diacylglycerol acyltransferase, *PDAT*—phospholipid; diacylglycerol acyltransferase, *DHAP*—dihydroxyacetone phosphate, *G3P*—glycerol 3-phosphate, *LPA*—lysophosphatidate, *PA*—phosphatidate, *DAG*—diacylglycerol, *PC*—phosphatidylcholine, *TAG*—triacylglycerol.

**Figure 3 plants-09-01135-f003:**
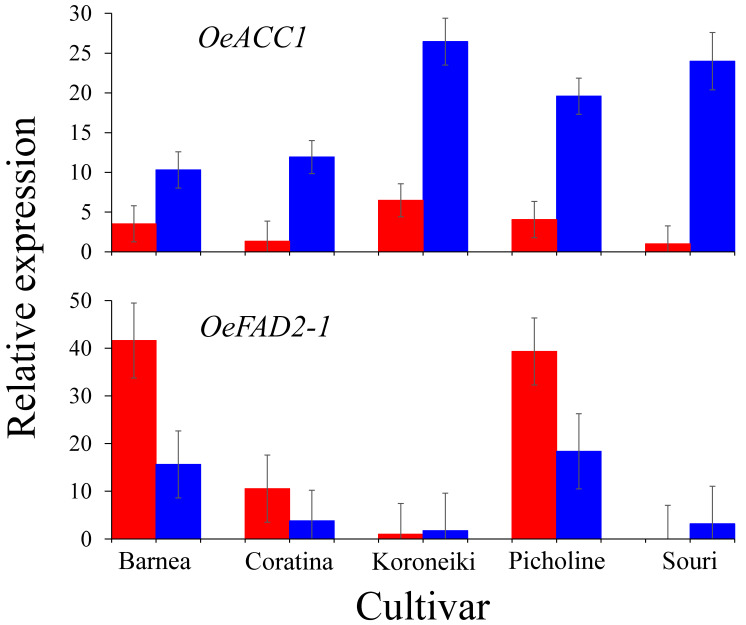
*OeACC1* and *OeFAD2-1* expression analyzed by RT-PCR at 146 DPA. Relative expression levels of *OeACC1* (above) and *OeFAD2-1* (below) in the five cultivars ‘Barnea’, ‘Coratina’, ‘Koroneiki’, ‘Picholine’ and ‘Souri’, in HT (red columns) and MT (blue columns) environments. Error bars represent confidence intervals (*p* < 0.05).

**Figure 4 plants-09-01135-f004:**
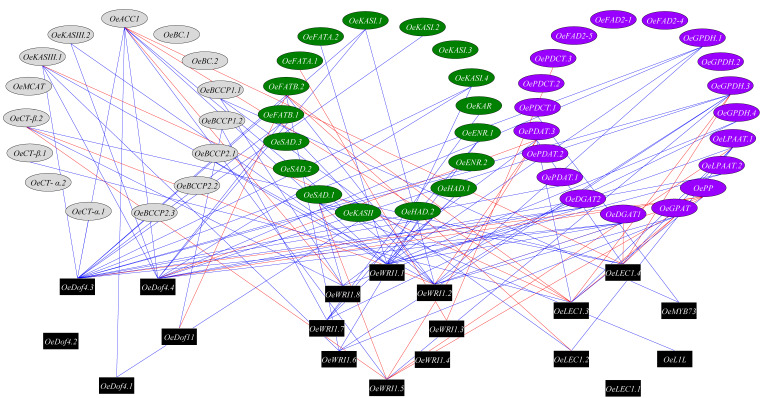
TF-oil biosynthesis network. Co-expressed network of the correlation among Transcription factors (colored in black) and Oil biosynthesis genes. The blue and red lines indicate positive and negative correlations between two genes, respectively. The oil biosynthesis genes are colored in green, gray and purple for plastid, cytosol and ER, respectively.

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
