# Peer review of "A High Temperature Environment Regulates the Olive Oil Biosynthesis Network"

_plants, 2020, doi:10.3390/plants9091135_

Round 1

Reviewer 1 Report

The submitted Manuscript ID plants-891617 and entitled " High temperature environment regulates olive oil biosynthesis network" by Yael Nissim et al. described the genome-wide transcriptional patterns in response to high temperatures in two cultivars with different sensitivity to heat stress. Overall the manuscript is well written and easy to follow. They identified several genes (HSP and oil biosynthesis related genes) that exhibited different transcriptional activities in the two cultivars. These findings would provide useful information for this research field.

I have several concerns below as some key information are missing in the current version.

Did you have biological repeats for RNA-seq? I didn't see this clearly explained in the method section.

L116, Here please give more information about the phenotype (quantity and quality) of the two cultivars. for example, oil content of fresh weight of each olive fruit? and L119, what do you mean by the oil quality? the chain length of fatty acid or number of double bonds?

L199: how to define "significantly"? Any statistical analysis here?

L268-269, This sentence is a repeat of Introduction L61-65. The repeated introductions of research background also happen in other places (Discussion), please double check.

L279-L280: hard to follow the meaning of this sentence.

L177 and for Figure 2, please define the meaning of the color bar. Is it the Z-score normalized FPKM values? Why genes with Z-score less than 0 were removed from the analysis as described in L263? I suggest the authors to explain the methodology and criteria more clearly in L177.

Figure 1, It's interesting that in Souri in HT, many HSP genes are induced in 83 DPA as indicated by the red color in the heatmap, then repressed (turning green) in 104 DPA, and up-regulated again in 146 DPA, totally different from their expression pattern in Barnea. Any explanations on their expressional pattern?

Figure 2. The author said the transcriptional patterns of oil-biosynthesis related genes are mostly similar in Souri and Barnea in L311, except for several genes. I suggest they also put the heat maps of genes in Barnea in Figure 2. That would be easy for readers to visualize the similar or difference of transcription patterns in the two cultivars.

Author Response

All responses are in a red font.

Reviewer 2 Report

This original research article written by Nissim et al. explored the molecular mechanism underlying the effect of a high temperature environment on oil quantity and quality in developing olive mesocarp. RNA-seq-based differential expression analysis revealed candidate genes in the triacylglycerol biosynthetic pathway, which are responsible for the quantity and quality of oil in two olive cultivars. RT-PCR experiments showed the expression levels of three key genes in five olive cultivars. Gene co-expression network analysis demonstrated that the expression levels of four transcription factors are tightly associated with those of the genes encoding glycerolipid biosynthetic enzymes. These results aid in development of solutions to deal with global warming.

Comments:

  1. Figure 2 (RNA-seq): Is this the result of the ‘Souri’ cultivar? I think the authors should show the result of the ‘Barnea’ cultivar as well, because the authors compared both cultivars and well discussed the results of the two cultivars in the discussion section. I couldn’t review these discussions because I couldn’t find the results of the expression pattern of the genes involved in the olive oil biosynthesis in the ‘Barnea’ cultivar.
  2. Figures 3 and S2 (RT-PCR): It would be better to describe when the mesocarp was sampled during the season for the RT-PCR experiment; 83, 104, or 146 DPA? Were the relative expression levels obtained from qRT-PCR consistent with the results from RNA-seq?

Minor comments:

  1. Figure 2: Z score is a normalization method by which each value is subtracted with mean of all the analyzed samples and divided by standard deviation of all the analyzed samples. So I think the heatmap of each gene should be colored in both green and red, but not only red (e.g. ACC1). Could you please explain which samples were used to calculate these Z scores? It might include the dataset of the ‘Barnea’ cultivar.
  2. Figures 2 and 4: In my knowledge, htACC (BCCP, BC, alpha-CT, beta-CT) and MCAT are localized in plastids to produce de novo fatty acids. hmACC (ACC1) is mainly localized in cytosol to produce very-long-chain fatty acids, and possibly also in plastids in monocot plants. So, it would be better to show these reactions under light green background.
  3. According to Figure 2, I would think, the expression levels of TAG synthesis genes (e.g. DGAT1) decreased under heat stress, which lead to a decrease in TAG content (oil quantity). It would be nice to discuss it as one of the “negative effect of a high temperature environment on oil quantity and quality”.
  4. Figure 4: This result demonstrates the gene co-expression network of olive oil biosynthesis very well, but it is not readable. For instance, it would be kind to show the gene names (nodes) with larger font size.

Author Response

All responses are in a red font.

Reviewer 3 Report

Overall it is a well-written manuscript with a good description of results. However, this manuscript needs revision before publication. Below are some of the shortcomings:

  1. The introduction section is not written appropriately. Authors should state the problem and what has been done, what is known particularly in response to high temp stress on oil synthesis in plants and in olive. There are several studies in different plant species already been published which should be cited. The introduction is written as a textbook type without stating the previous studies related to oil synthesis and high temp effect. This needs to be revised.
  2. Section 1.1: Why the focus is only on HSP70 and why not others such as HSP 90 and small HSPs?
  3. Section 1.2: Authors should summarize this in short and provide appropriate references. The oil biosynthesis pathways are already well described and no need for this elaborate description until unless they are different in olive.
  4. Section 1.3: this should be merged with the above section as it is the part of oil biosynthesis pathway.
  5. Authors stated the HT effect on oil quantity and quality. What is meant by quality? change in FA composition/profile? This should be stated in the introduction with appropriate references. 
  6. No data on oil quality was presented. Did the heat stress affected the Fatty acids composition and profile. Authors should study that to see the effect on oil quality and whether changes in the gene expression related to FA saturation in fact did affect the quality? This is critical a critical gap in the data presented.
  7. Figure 3 and supplementary figure 2: Authors studied the Transcriptome analysis for only for two cultivars- Barnea and Souri. Why authors are comparing 5 cultivars here? Any specific reason?
  8. Figure 4 data: Since it is an in-silico analysis and does not add much value to the results. It should be presented as supplementary data

Author Response

All responses are in a red font.

Round 2

Reviewer 2 Report

The manuscript has been significantly improved. The authors answered all the comment from the reviewer 2.

Figure 2: The authors showed the result of the ‘Barnea’ cultivar. Thank you very much.

Figures3 and S2: The authors showed the sampled date, and also added the comparison between the RNA-seq and the RT-PCR results in the discussion.

In Abstract: Now I agree that OePDCT and OeFAD2 showed cultivar dependent expression patterns.

Reviewer 3 Report

The authors have satisfactorily addressed all of my concerns. The manuscript is acceptable.